# Characterization of Luminescent Down-Shifting Spectral Conversion Effects on Silicon Solar Cells with Various Combinations of Eu-Doped Phosphors

**DOI:** 10.3390/ma15020452

**Published:** 2022-01-07

**Authors:** Wen-Jeng Ho, Jheng-Jie Liu, Bo-Xun Ke

**Affiliations:** Department of Electro-Optical Engineering, National Taipei University of Technology, 1, Sec. 3, Zhongxiao E. Rd., Taipei 10608, Taiwan; jjliu@mail.ntut.edu.tw (J.-J.L.); karta324222@gmail.com (B.-X.K.)

**Keywords:** Eu-doped, luminescent down-shifting, phosphors, solar cell, spectral conversion, spin coating

## Abstract

Luminescent down-shifting (LDS) spectral conversion is a feasible approach to enhancing the short-wavelength response of single junction solar cells. This paper presents the optical and electrical characteristics of LDS spectral conversion layers containing a single species or two species of Eu-doped phosphors applied to the front surface of silicon solar cells via spin-on coating. The chemical composition, surface morphology, and fluorescence emission of the LDS layers were respectively characterized using energy-dispersive X-ray analysis, optical imaging, and photoluminescence measurements. We also examined the LDS effects of various phosphors on silicon solar cells in terms of optical reflectance and external quantum efficiency. Finally, we examined the LDS effects of the phosphors on photovoltaic performance by measuring photovoltaic current density–voltage characteristics using an air-mass 1.5 global solar simulator. Compared to the control cell, the application of a single phosphor enhanced efficiency by 17.39% (from 11.14% to 13.07%), whereas the application of two different phosphors enhanced efficiency by 31.63% (from 11.14% to 14.66%).

## 1. Introduction

It has been estimated that by 2035, renewable and green energy sources will account for more than half of global electricity generation [1,2,3]. The fact that photovoltaics provides nearly unlimited access to clean solar energy ranks it among the renewable energies with the greatest potential [4,5]. Solar cells are semiconductor devices that convert solar radiation directly into electricity via the photovoltaic effect [6]. Crystalline silicon (C-Si) solar cells are currently the mainstay of the photovoltaic industry, owing to the abundance of the constitutive materials and the maturity of the underlying techniques. These devices currently account for 80% of the photovoltaic devices in use worldwide [7]. Unfortunately, the energy conversion efficiency of C-Si solar cells is limited by the fact that they respond only to photons within a narrow region of the solar spectrum, such that much of the photon energy exceeds the bandgap of the silicon semiconductor.

Considerable effort has gone into developing methods by which to increase the conversion efficiency of C-Si solar cells, including the formation of pyramidal surface structures and the application of anti-reflective coatings [8,9,10,11,12]. Nonetheless, the conversion efficiency of C-Si solar cells at shorter wavelengths (UV-blue band) remains low due to three key issues: (1) lower responsivity of C-Si in the UV-blue wavelength range; (2) loss of higher energy incident photons within a short distance of the surface due to photo-carrier recombination caused by surface defects; (3) energy loss via thermalization when high energy photons generate carriers with excessive energy [13,14,15].

Luminescent down-shifting (LDS) and down conversion (DC) are new approaches to harvesting high-energy incident photons from the sun through the conversion of high-energy photons into multiple photons of low energy to facilitate conversion into electricity [16,17,18,19,20]. LDS and DC spectral conversion layers improve the responsivity of C-Si solar cells operating at shorter wavelengths while reducing recombination loss and thermalization loss. Lanthanides are used extensively in spectral conversion due to their rich energy-level configuration, which facilitates photon management [21,22]. Europium-doped (Eu-doped) phosphors are ideally suited to LDS, owing to their high luminescent quantum yield and large Stokes shift [23,24,25]. Researchers have demonstrated the efficacy of adding to solar cells an LDS layer with single-species or multi-species Eu-doped phosphors [26,27,28,29]. However, there has been relatively little research on the fabrication of solar cells using two such layers with two different phosphors [30,31,32,33].

In the present study, we conducted systemic analysis of C-Si solar cells with coated with LDS layers, comprising either an SiO_2_ layer containing a single phosphor applied via single-stage spin coating or two SiO_2_ layers containing different phosphors applied via multi-stage spin coating. The chemical composition, surface morphology, and fluorescence emission of the phosphor layers were respectively characterized using energy-dispersive X-ray analysis, optical imaging (OM), and photoluminescence (PL) measurements. We also compared the LDS effects of various phosphors on reflectance and external quantum efficiency (*EQE*). We conducted photovoltaic current density vs. voltage (J–V) measurements to evaluate the effects of these coatings on solar cell efficiency. We also compared the solar cell efficiency of a single coating with one phosphor type with that of two coatings containing two phosphor types.

## 2. Material and Methods

### 2.1. Characterization of SiO_2_ LDS Layers Containing Phosphors

#### 2.1.1. Single Coating with Single Phosphor Particles

In this study, three Eu-doped phosphors (referred to as Phosphor-E, -F, and -G) were assessed in terms of LDS spectral conversion from ultraviolet (UV) to visible (VIS) wavelength bands. The phosphors were obtained from InteMatix (Fremont, CA, USA). Figure 1a presents a schematic diagram showing a silicon substrate (or silicon solar cell) on which was deposited an LDS spectral conversion layer of SiO_2_ containing a single phosphor. Into a solution containing 1.94 g silicate was mixed 0.06 g (3 wt %) of a single phosphor in powdered form (the particle size from 5 to 20 μm). Then, the LDS layer was created by coating a clean silicon substrate with this solution via spin-on processing at 3500 rpm for 55 s, before being baked at 210 °C for 110 s under an air atmosphere. Note that the mixed solution was applied to the sample drop-wise and held for 5 s prior to spinning.

#### 2.1.2. Double-Coatings Using Two Types of Phosphor Particles

As shown in Figure 1b, we fabricated double-layer LDS samples by applying a second SiO_2_ layer containing a different phosphor over the first layer using the same spin-on coating process. Before applying the second coating, the samples underwent ultrasonic cleaning in acetone and methanol solutions for 5 min followed by rinsing in DI water for 4 min before being dried under ambient N_2_ for 2 min. Note that in both coating solutions, the concentration of phosphors was 3 wt %. The resulting 1st-layer/2nd-layer configuration is hereafter described according to the constituent phosphor particles: Phosphor-E/-F, Phosphor-F/-G, and Phosphor-G/-E.

The morphology and chemical makeup of samples with a phosphor layer were examined via optical microscopy (OM) and scanning electron microscopy/energy-dispersive spectroscopy (SEM/EDS; Hitachi S-47000, Hitachi High-Tech Fielding Corporation, Tokyo, Japan). The fluorescence emission of the phosphor layer was characterized by obtaining PL measurements at room temperature. The PL (Spex Fluorolog-3, Jobin Yvon Instrument S. A. Inc., Sunnyvale, CA, USA) and optical reflectance spectra were used to reveal the LDS effects of the phosphor layer(s). The reflectance of samples (with and without phosphor particles in the SiO_2_ layer) was characterized by an UV/VIS/NIR spectrophotometer (PerkinElmer Lambda 35, Waltham, MA, USA).

### 2.2. Characterization of Solar Cells with a Single Layer or Dual Layers of SiO_2_ Containing Phosphors

The starting material for the solar cell devices was a p-type (100) C-Si wafer (550 μm 10 Ω-cm). The wafer was first cut into samples measuring 1 × 1 cm^2^, which were then subjected to the standard RCA cleaning process. An n^+^-Si layer (emitter, 0.35 μm) was created on the front-surface of C-Si via spin processing using a liquid phosphorous source (Phosphorofilm, Emulstione Co., Washington, NJ, USA) followed by heating in a rapid thermal annealing (RTA) chamber at 910 °C for 100 s under ambient N_2_. The diffused oxide layer remaining on the surface of the Si sample was removed by a BOE solution. Four-point probe measurement showed that the sheet resistance of the n^+^-Si emitter-layer was roughly 60 Ω/cm^2^. SIMS profiling revealed that the surface phosphorus concentration was roughly 1.5 × 10^20^ cm^−3^. Clean diffused samples underwent isolation, etching to a depth of 0.85 μm via photolithography using an etching solution of HNO_3_:HF:H_2_O (1:1:2) for 35 s, which resulted in individual cells measuring 4.1 × 4.1 mm^2^. A P-electrode and N-electrode were produced as Ohmic contacts by depositing 500 nm-thick Al film on the back side and a Ti (25 nm)/Al (350 nm) on the top surface using e-beam evaporation and a lift-off process. Then, the samples underwent annealing in an RTA chamber under ambient N_2_ at 445 °C for 23 min to optimize metal/semiconductor contact characteristics, thereby completing the bare-type Si solar cells.

To investigate the effects of LDS on the performance of C-Si solar cells, we coated bare-type C-Si solar cells with a SiO_2_ layer containing phosphors (Phosphor-E, or -F, or -G) at a concentration of 3 wt % via spin coating (hereafter referred to as single-phosphor/single-coating), as shown in Figure 1a. We assessed broadband LDS effects by applying a second LDS layer over the first layer (hereafter referred to as two-phosphors/double-coating), as shown in Figure 1b. Note that different phosphors were used in the first and second coatings (E/F, F/G, or G/E), but the phosphor concentrations in the two layers were the same (3 wt %). Then, we examined the optical and electrical characteristics of the resulting luminescent down-shifting spectral conversion coatings in terms of optical reflectance, *EQE* response at wavelengths between 300 and 1000 nm (LSQE-R3015, Enli Technology Co., Kaohsiung, Taiwan), and J–V curves under AM 1.5 G solar simulation. The XES-151S solar simulator (San-Ei Electric Co., Ltd., Osaka, Japan) used in this work was calibrated using a PVM-894 silicon reference cell (PV Measurements Inc., Boulder, CO, USA) certified by the National Renewable Energy Laboratory before tests. All presented data were averaged from three measurements.

## 3. Results and Discussion

### 3.1. Characteristics of LDS Layers Containing Phosphors

Figure 2 presents optical images and the particle size distribution of phosphor particles in samples with a single-phosphor SiO_2_ layer containing (a) Phosphor-E, (b) Phosphor-F, and (c) Phosphor-G. Particle size profiles were derived from the optical images using Image-J software. Due to similarities in the spin-coating coating parameters, the average diameter (14 μm) and surface coverage (14%) of all samples were roughly the same, regardless of the constituent phosphors. Figure 2d presents SEM images (top and side views) of silicon wafers coated with SiO_2_ and phosphor particles. SEM revealed that the average diameter of the phosphor particles closely matched the OM results (roughly 14 μm). Figure 3 presents the surface morphology and particle-size distribution of samples with two phosphors. The average particle diameter (17 μm) and surface coverage (21%) on the double-coating samples were both higher than on the single-coating samples due to the fact that the empty areas left in the first coating were filled with particles applied during the second coating, and a few particles from the first and second layers overlapped. Figure 2 and Figure 3 present Gaussian fittings to enable a statistical comparison.

Energy-dispersive spectroscopy (EDS) is used in the element analysis of solid samples to characterize the interaction between the X-ray excitation source and sample. Each element within the sample has a unique atomic structure, which produces a unique set of peaks in the EDS spectrum. Figure 4 displays the EDS spectra of SiO_2_ layers containing different phosphors: (a) Phosphor-E, (b) Phosphor-F, and (c) Phosphor-G. The composition of all phosphor samples was mainly Si, O, Ba, and Sr, with a small quantity of Eu and other elements (F, Ti, Mn, Mg, and Cl). The chemical formulas were as follows: Phosphor-E ([(Sr_0.05_Ba_0.95_)_0.98_Eu_0.02_]_2_SiO_3.9_F_0.1_), Phosphor-F ([(Sr_0.7_Ba_0.3_)_0.98_Eu_0.02_]_2_SiO_3.9_F_0.1_), and Phosphor-G ((Sr_0.9_Ba_0.1_)_3_SiO_5_:Eu), based on information obtained from InteMatix products information.

Figure 5 displays the PL spectra and color optical emission images of samples with a SiO_2_ layer containing a single phosphor: (a) Phosphor-E, (b) Phosphor-F, and (c) Phosphor-G. The optical images were obtained from samples under blue light illumination. The PL emission peak (*λ*_PL_) and color were as follows: Phosphor-E (*λ*_PL_ = 514.7 nm and cyan), Phosphor-F (*λ*_PL_ = 546.6 nm and green), Phosphor-G (*λ*_PL_ = 603.3 nm and orange).

Figure 6 displays the PL spectra and color optical emission images of samples with 1st-SiO_2_ (bottom)/2nd-SiO_2_ (top) layers comprising various combinations of two phosphors: (a) Phosphor-E/Phosphor-F, or (b) Phosphor-F/Phosphor-G, or (c) Phosphor-G/Phosphor-E. The PL emission-peak and color were as follows: E/F (*λ*_PL_ = 545.5 and green), F/G (*λ*_PL_ = 592.6 nm and orange), and G/E (*λ*_PL_ = 514.3 nm and cyan). Overall, the PL intensity of samples with two phosphors exceeded that of single-phosphor samples (>1.5 times). In addition, the peak PL emission wavelength of dual-phosphor combinations shifted toward the emission wavelength of the 2nd-layer containing Eu-doped phosphor, with a corresponding shift in color toward that of the 2nd layer. Wideband performance was particularly evident in the F/G sample.

### 3.2. Performance of Silicon Solar Cells Coated with LDS Layers Comprising One or Two Phosphors

The LDS effect on the efficiency of C-Si solar cells was characterized in terms of optical reflectance, *EQE*, and J–V characteristics under AM 1.5 G illumination. These values were first obtained from bare cells and cells with a single SiO_2_ layer for use as a reference. The open-circuit voltage (*V*_oc_), short-circuit current-density (*J*_sc_), and conversion efficiency (*η*) of bare cells were as follows: *V*_oc_ = 541.3 mV, *J_s_*_c_ = 26.91 mA/cm^2^, and *η* = 11.14%. Figure 7a presents the measured reflectance from the three types of solar cells examined in this experiment: bare cells, cells coated with a layer of SiO_2_, and cells coated with a layer of SiO_2_ containing phosphors. Figure 7a also shows the simulated reflectance of bare Si/SiO_2_. Note that the SiO_2_ layer was created using a single spin-coating operation using silicate solution. In simulations using TFCalc^TM^ simulation software, the thickness of the SiO_2_ layer was estimated at 245 nm; however, the measured thickness of SiO_2_ in actual samples was roughly 240 nm, as confirmed in side-view SEM images (inset in Figure 7a). The reflectance of the cell with a SiO_2_ layer was lower than that of the bare cell across the entire range of wavelengths, which was due to the anti-reflective effects of the SiO_2_ film. The reflectance of cells with phosphor particles was even lower than that of cells with only a SiO_2_ layer over a wavelength range of 350 to 410 nm (owing to the absorption of incident light by the phosphors) and 550 to 900 nm (owing to the forward scattering of incident photons). Figure 7b presents the measured reflectance and simulated reflectance of cells with double-coatings. In simulations, the thickness of samples with a double coating of SiO_2_ was roughly 350 nm; however, the thickness of actual samples was 357 nm, as confirmed in side-view SEM images (inset in Figure 7b). The overall reflectance of samples with a double-coating of SiO_2_ was lower than that of samples with a single-coating, and the wavelength corresponding to the lowest reflectance was red-shifted from 500 to 700 nm. The reflectance values of cells with a double coating of SiO_2_ containing phosphors was lower than the single layer equivalents over a limited wavelength range (350 to 410 nm) as well as samples with a double-coating of SiO_2_ over the full wavelength range (350–1000 nm). The G-/E-phosphor combination presented the lowest overall reflectance. As shown in Table 1, the average weighted reflectance (*R*_W_) was derived for two wavelength ranges (350–450 nm and 350–1000 nm). *R*_W_ values were used to characterize the LDS effects induced by phosphors (single-phosphor and two-phosphor). Based on the reflectance results, we expected that samples with a double coating of SiO_2_ containing two different phosphors would present the highest *EQE* response.

Figure 8a displays the *EQE* spectra of a bare cell, a cell coated with a layer of SiO_2_, and cells coated with a layer of SiO_2_ containing various phosphors. The *EQE* of cells with a double coating of SiO_2_ containing phosphors was higher than that of the cell with only SiO_2_, except over a wavelength range of 460 to 630 nm, owing to LDS effects and forward scattering caused by the phosphors. The *EQE* of the sample with Phosphor-G was higher than that of the samples with Phosphor-F or Phosphor-E over a wavelength range of 350 to 450 nm. As shown in Table 1, we calculated the average external quantum efficiency (*EQE_W_*) for wavelength ranges of 350–450 nm and 350–1000 nm. The *EQE* results are in good agreement with those of optical reflectance and PL results, due to the fact that the photons emitted by Phosphor-G and Phosphor-F fall within the range of wavelengths to which Si devices are highly responsive. Figure 8b presents the *EQE* spectra of the cell with two phosphors. The overall *EQE* value of double-coated samples exceeded that of single-coated samples, and the wavelength corresponding to the highest *EQE* was red shifted from 550 to 630 nm. Similarly, the *EQE* values of the cells with two phosphors exceeded those of cells with a single coating over a wavelength range of 350 to 410 nm. The highest *EQE* was obtained from the G/E sample. Table 1 also lists the *EQE_W_* of the double-layer cells. We expected that samples with a double coating of SiO_2_ containing two phosphors (G/E configuration) would present the highest photovoltaic performance. Table 1 lists the integrated *J*_sc_ values from the EQE spectra to enable a comparison of LDS and anti-reflection effects. To further clarify the relationship, we calculated the *EQE_W_* as follows:$$EQE_{W}=\frac{\int_{350nm}^{1000nm}EQE(\lambda )\phi (\lambda )d\lambda }{\int_{350nm}^{1000nm}\phi (\lambda )d\lambda }\times 100\%$$
where *EQE(λ)* is the *EQE* at a given *λ* (wavelength), and *ϕ(λ*) is the photon flux of AM 1.5 G at that *λ*.

Figure 9a presents the J–V curves of a bare cell, a cell with a layer of SiO_2_, and cells with a layer of SiO_2_ containing Eu-doped phosphors obtained under AM 1.5 G solar simulation. The *V*_oc_ (544.8 mV), *J*_sc_ (29.2 mA/cm^2^), and *η* (12.28%) of the cell coated with a SiO_2_ layer exceeded those of the bare cell (541.3 mV, 26.91 mA/cm^2^, 11.14%), owing to the passivation and anti-reflection effects of the SiO_2_ layer. The *J*_sc_ (30.04–30.97 mA/cm^2^) and η (12.82–13.07%) of the cells with a single phosphor were higher than those of the cell with only SiO_2_, owing to LDS effects and forward scattering. The cell with Phosphor-G presented *J*_sc_ (30.97 mA/cm^2^) and η (13.07%) values that were significantly higher than those of the cell with Phosphor-F (30.26 mA/cm^2^, 12.93%) or Phosphor-E (30.04 mA/cm^2^, 12.82%). Table 2 lists the conversion efficiency enhancement (Δ*η*_DS_) of cells with an LDS layer (SiO_2_: phosphors, one-phosphor), compared to the cell with a single coating of pure SiO_2_. Figure 9b presents the photovoltaic J–V curves of cells with two-phosphor combinations. The *J*_sc_ and *η* values of dual-phosphor dual-coating samples exceeded those of their single-phosphor single-coating counterparts. The *J*_sc_ enhancement of dual-layer phosphor cells (Δ*J*_sc_ = 19.81–28.91%) and η (Δ*η* = 22.03–31.63%) exceeded those of their single-layer counterparts (Δ*J*_sc_ = 11.63–15.09% and Δ*η* = 15.16–17.39%), which was comparable to those of the bare reference cell. The Phosphor-G/E combination achieved the highest *J*_sc_ and *η* values overall. Table 2 lists the Δ*η*_DS_ of dual-phosphor dual-coating samples compared to the cell with double-coating of pure SiO_2_ layers. Table 2 also summarizes the photovoltaic performance of all evaluated cells.

## 4. Conclusions

This study examined the optical and electrical effects of a luminescent down-shifting (LDS) layer containing a single species or two species of Eu-doped phosphors applied via spin coating. PL, optical reflectance, and *EQE* measurements revealed the LDS effects of the phosphors. The highest conversion efficiency (13.07%) was obtained in solar cells with Phosphor-G, which was due to the high responsivity of the silicon to LDS photons. The combination of two phosphors had more pronounced effects on broadband LDS emissions than did the single coatings containing a single phosphor. The solar cell with a Phosphor-G/Phosphor-E combination yielded the highest short-circuit current density (34.69 mA/cm^2^) and conversion efficiency (14.66%) corresponding to a 31.63% improvement over that of bare Si solar cells.

## Figures and Tables

**Figure 1 materials-15-00452-f001:**
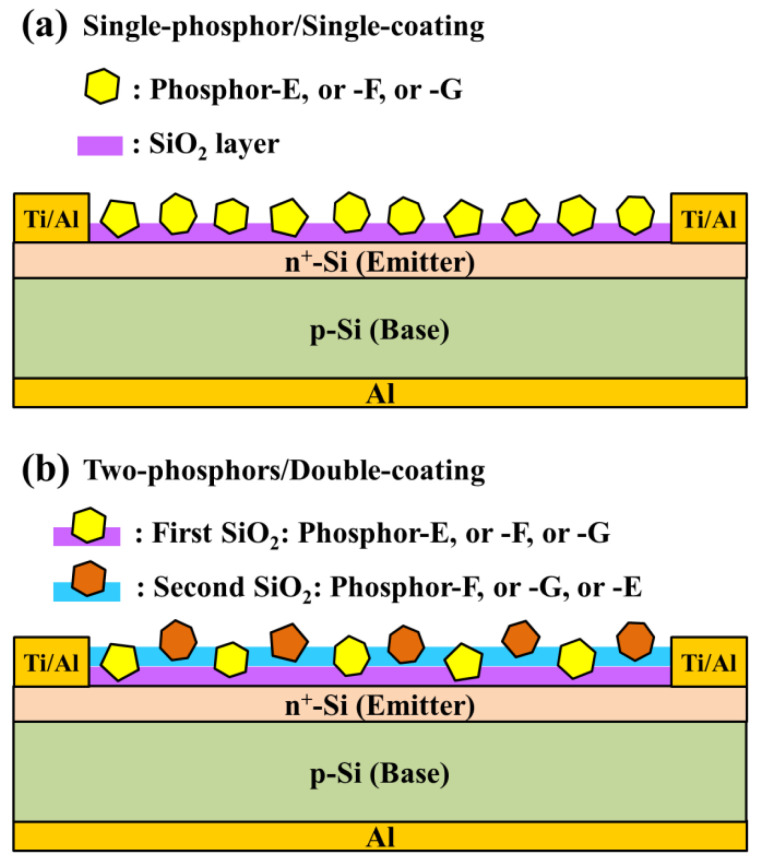
Schematic diagram of the sample with (**a**) single coating of SiO_2_ containing a single phosphor, and (**b**) a double coating of SiO_2_ containing two different phosphors.

**Figure 2 materials-15-00452-f002:**
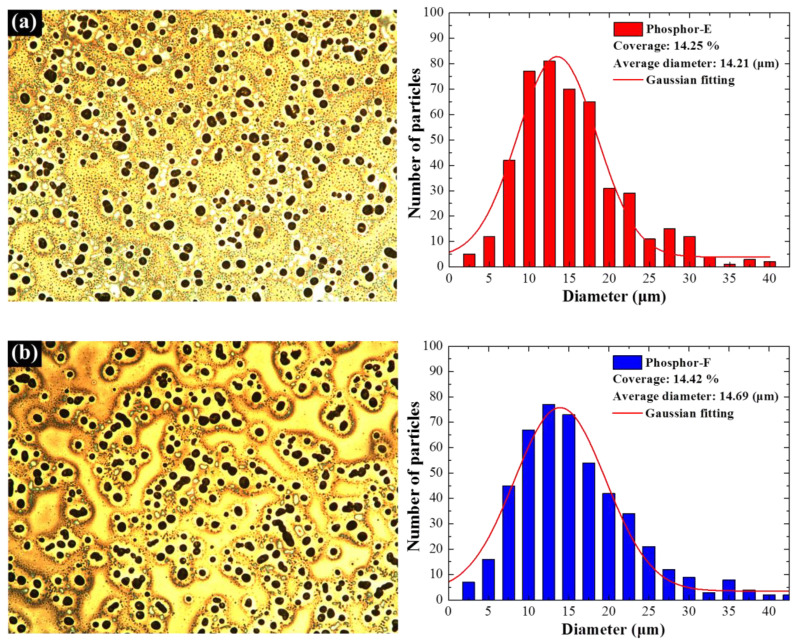
Optical images and particle-size distributions of single-phosphor samples: (**a**) Phosphor-E, (**b**) Phosphor-F, and (**c**) Phosphor-G; (**d**) SEM images (top and side views) of the silicon wafer coated with SiO_2_ and phosphor particles.

**Figure 3 materials-15-00452-f003:**
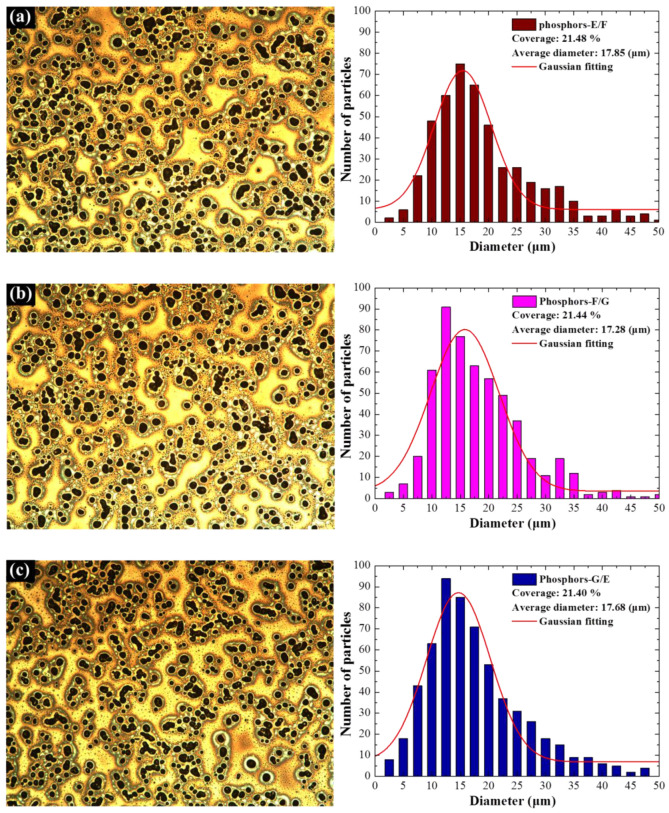
Optical images and particle-size distribution of samples with dual-coatings of phosphors: (**a**) E/F; (**b**) F/G; and (**c**) G/E.

**Figure 4 materials-15-00452-f004:**
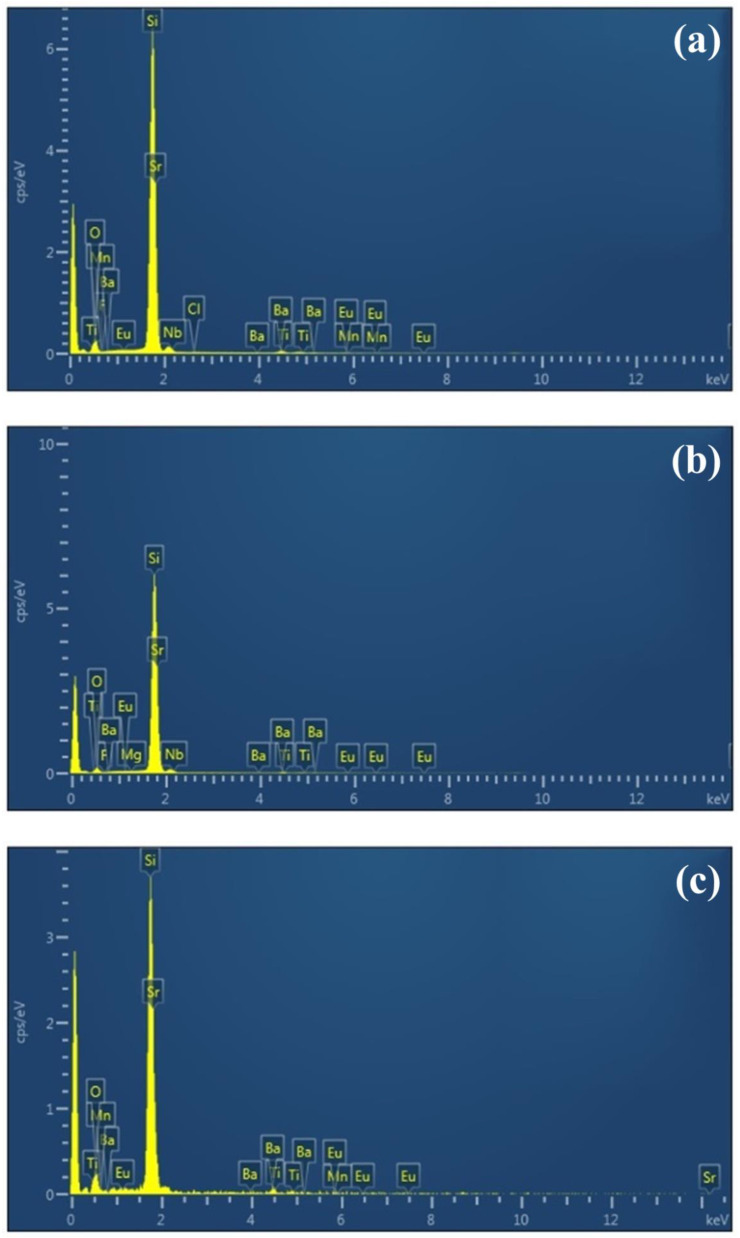
EDS spectra of silicon substrates deposited a SiO_2_ layer containing a single phosphor: (**a**) Phosphor-E; (**b**) Phosphor-F; and (**c**) Phosphor-G.

**Figure 5 materials-15-00452-f005:**
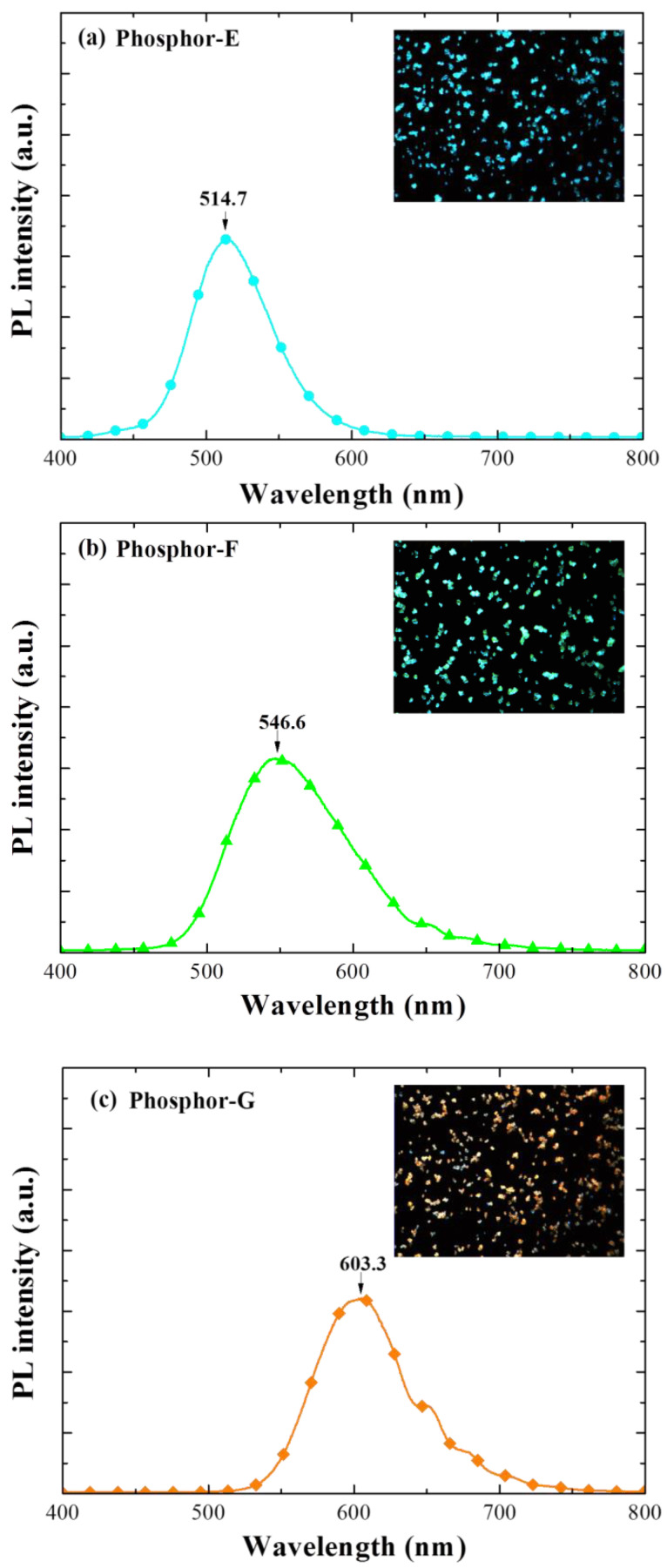
PL spectra and color optical emission images, for the samples with a SiO_2_ layer containing single phosphors: (**a**) Phosphor-E, (**b**) Phosphor-F, and (**c**) Phosphor-G.

**Figure 6 materials-15-00452-f006:**
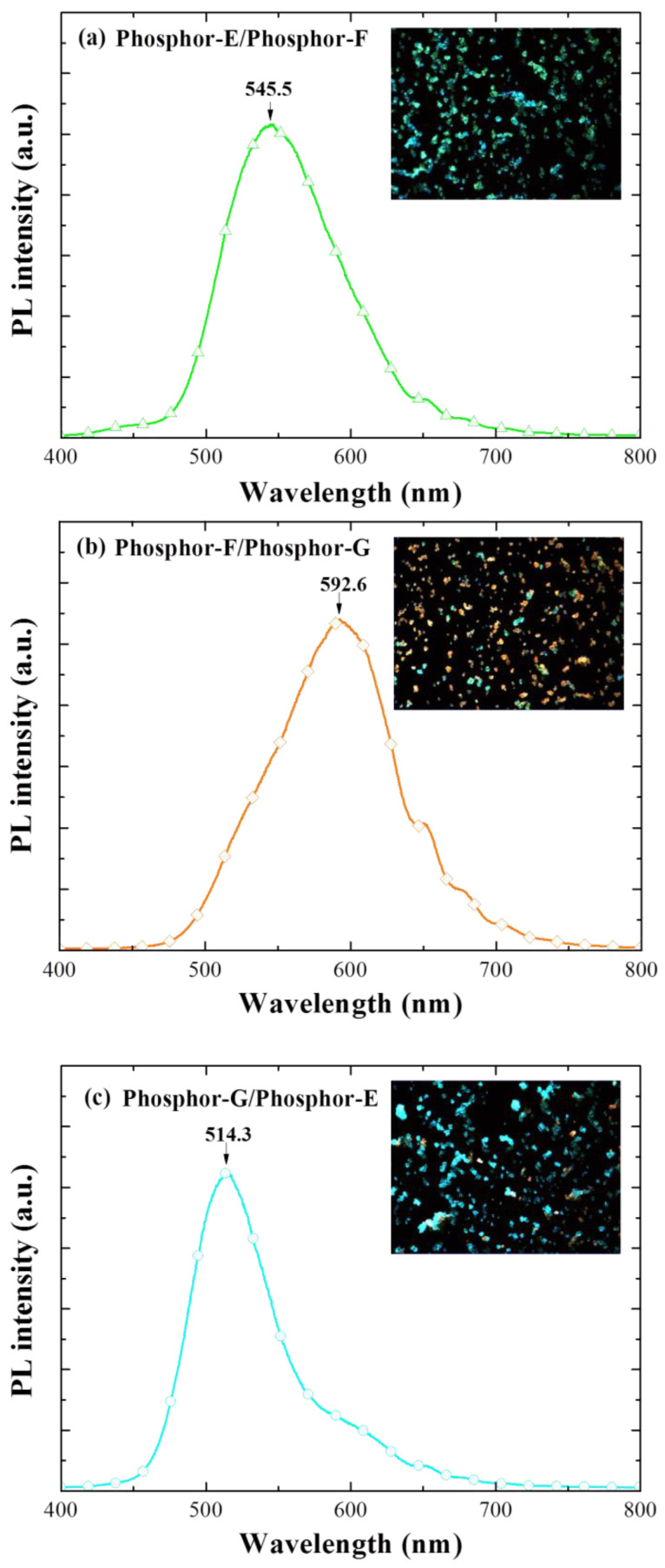
PL spectra of samples with bottom/top LDS layers containing two phosphors: (**a**) E/F; (**b**) F/G; (**c**) G/E.

**Figure 7 materials-15-00452-f007:**
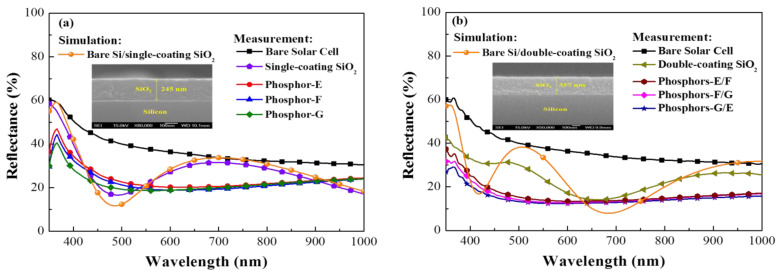
Measured and simulated reflectance of all cells in this study: (**a**) single-coating with single phosphor; (**b**) double-coating with two phosphors.

**Figure 8 materials-15-00452-f008:**
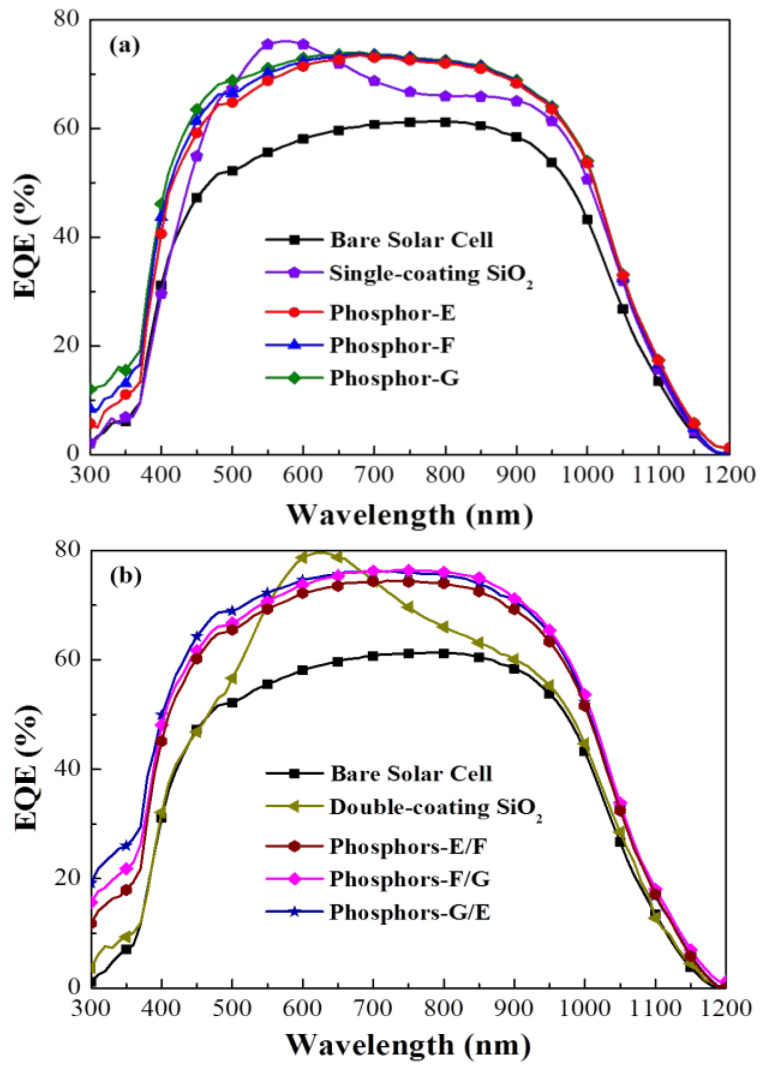
*EQE* spectra of proposed silicon solar cells: (**a**) single coating; (**b**) double coating.

**Figure 9 materials-15-00452-f009:**
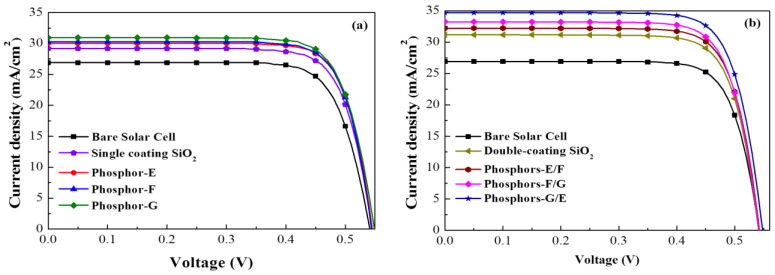
J–V curves of silicon solar cells: (**a**) single coating; (**b**) double coating, under AM 1.5 G simulation.

**Table 1 materials-15-00452-t001:** Average weighted reflectance (*R*_W_) and average weighted external quantum efficiency (*EQE_W_*) calculated for wavelength ranges of 350–450 nm and 350–1000 nm. Integrated *J*_sc_ values calculated from *EQE* spectra (350–1000 nm).

Sample	*R*_W_ (%) @ 350–450 nm	*R*_W_ (%) @ 350–1000 nm	*EQE_W_* (%) @ 350–450 nm	*EQE_W_* (%) @ 350–1000 nm	*Cal. J*_sc_ (mA/cm^2^)
Bare solar cell	49.84	37.35	32.20	52.93	22.29
Single-coating SiO_2_	36.18	26.74	33.86	62.47	26.19
SiO_2_: Phosphor-E	33.48	23.43	41.68	64.30	27.04
SiO_2_: Phosphor-F	31.27	22.14	43.99	65.26	27.29
SiO_2_: Phosphor-G	27.78	21.45	46.40	66.21	27.66
Double-coating SiO_2_	33.45	23.62	35.96	64.08	26.54
Phosphor-E/Phosphor-F	24.80	16.16	45.39	65.59	27.73
Phosphor-F/Phosphor-G	22.47	15.31	47.74	67.44	28.28
Phosphor-G/Phosphor-E	20.11	14.42	50.34	68.17	28.67

**Table 2 materials-15-00452-t002:** Photovoltaic performance of bare cell, cell with SiO_2_ layer, and cells with phosphors.

Sample	*V*_oc_ (mV)	*J*_sc_ (mA/cm^2^)	FF (%)	*η* (%)	Δ*J*_sc_ (%)	Δ*η* (%)	Δ*η*_DS_ (%)
Bare solar (SC)	541.3	26.91	76.45	11.14	--	--	--
Single-coating SiO_2_	544.8	29.20	77.22	12.28	8.51	10.31	--
SiO_2_: Phosphor-E	546.8	30.04	78.07	12.82	11.63	15.16	4.40
SiO_2_: Phosphor-F	547.1	30.26	78.08	12.93	12.45	16.08	5.29
SiO_2_: Phosphor-G	548.1	30.97	77.01	13.07	15.09	17.39	6.43
Double-coating SiO_2_	542.6	31.18	77.22	13.06	15.87	17.32	--
Phosphor-E/Phosphor-F	543.4	32.24	77.57	13.59	19.81	22.03	4.06
Phosphor-F/Phosphor-G	544.4	33.25	77.28	13.99	23.56	25.62	7.12
Phosphor-G/Phosphor-E	547.4	34.69	77.19	14.66	28.91	31.63	12.25

## Data Availability

Not applicable.

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
