# Peer review of "Characterization of Luminescent Down-Shifting Spectral Conversion Effects on Silicon Solar Cells with Various Combinations of Eu-Doped Phosphors"

_materials, 2022, doi:10.3390/ma15020452_

Round 1

Reviewer 1 Report

  1. The authors should provide SEM images of the wafers with coated SiO2 and DLS powders. Why the use of microscale particles reduced the reflection?
  2. I somehow doubt that the improvement in solar cell efficiency was due to the reduction in reflection rather than DLS effect. The authors must give detailed characterization to make theri results clear.

Author Response

The authors would like to express their sincere appreciation to the Editors and Reviewers for useful comments. The reviewers’ comments are taken into account and mentioned issues have been addressed for significant improvement in the revised manuscript. In keeping with the comments of the Reviewer #1, Reviewer #2 and Reviewer #3, we have made some revisions by marking with the blue color and underline for the Reviewer #1, the red color and underline for the Reviewer #2, and the green color and underline for the Reviewer #3, respectively, at exact locations on the revised manuscript. Besides, the manuscript has been edited by Michael D. Ash, a Canadian (blue color without underline). The revised manuscript was edited for proper English language, grammar, punctuation, spelling, and overall style by more of the high qualified native English language editors at Acceptediting Editing (Please find Supplementary Material for Review: English Editing Certificate).

Reviewer #1:

Comments and Suggestions for Authors

  1. The authors should provide SEM images of the wafers with coated SiO2 and DLS powders. Why the use of microscale particles reduced the reflection?

Ans:

  1. Thanks for your useful comments.
  2. Top-view and side-view SEM images of the wafers with coated SiO2 and LDS powders were added in Fig. 2(d), on the revised manuscript.
  3. The reflectance reduction in short wavelength range (350 to 410 nm) was due to the absorption of incident light by the Eu-doped phosphors, however, the reflectance reduction in 550-900 nm wavelength range was due to the forward scattering of incident photons by the Eu-doped phosphor particles.

We revised as:

Figure 2(d) presents SEM images (top and side views) of silicon wafers coated with SiO2 and phosphor particle. SEM revealed that the average diameter of the phosphor particles closely matched the OM results (roughly 14 μm)”, on the revised manuscript.

  1. I somehow doubt that the improvement in solar cell efficiency was due to the reduction in reflection rather than DLS effect. The authors must give detailed characterization to make theri results clear.

Ans:

  1. Thanks for your useful comments.
  2. In Table 2, we added the conversion efficiency enhancement of the cell with an LDS layer (SiO2: Eu-doped phosphors, one-spices), compared to the cell coated with a single-layer of pure SiO2. The enhancement due to the reduction in reflection may be neglect because both cells have the same SiO2. Therefore, the enhancements from 4.40% to 6.43% were obtained due to using one-species Eu-doped phosphors.
  3. Similarly, In Table 2, we also added the conversion efficiency enhancement of the cell with two LDS layers (bottom LDS layer-the first coating/top LDS layer-the second coating) with different Eu-phosphors in the coating layer, compared to the cell with double-coating of pure SiO2 The enhancements from 4.06% to 12.25% were obtained due to using two-species Eu-doped phosphors combinations, which is higher than that of the cell with one-species Eu-doped phosphors.

We revised as:

Table 2 lists the conversion efficiency enhancement (ΔηDS) of cells with an LDS layer (SiO2: phosphors, one-phosphor), compared to the cell with a single coating of pure SiO2”, and “Table 2 lists the ΔηDS of dual-phosphor dual-coating samples, compared to the cell with double-coating of pure SiO2 layers. Table 2 also summarizes the photovoltaic performance of all evaluated cells”, on the revised manuscript. 

Reviewer 2 Report

Authors prepared Eu-related compounds for luminescent shift.

Since they purchased those chemicals from a company. They would like to discuss over experimental conditions rather than chemical themselves. I strongly suggest they add more introduction to condition-related luminescent shift such as double-sided silica etc.

Currently, they discussed Eu-related compounds, where they did not design nor synthesize those chemicals for their purpose. For example, I would like to know what I can expect from regular reference chemical on silica-based substrate. I would like to know what the variables in the experiments are. If any functional groups or moiety of chemicals are important, they want to discuss over the moiety or functional groups on luminescent data. Currently, they just showed and summarized the results without approaches for each chemical’s properties. For example, they used tables 1 and 2 to summarize the conditions, but in the text, I do not see what they want to compare, analyze, and discuss. They just summarized the situations. Please give more detailed explanation for the purpose of the experiments.

Thanks.

Author Response

The authors would like to express their sincere appreciation to the Editors and Reviewers for useful comments. The reviewers’ comments are taken into account and mentioned issues have been addressed for significant improvement in the revised manuscript. In keeping with the comments of the Reviewer #1, Reviewer #2 and Reviewer #3, we have made some revisions by marking with the blue color and underline for the Reviewer #1, the red color and underline for the Reviewer #2, and the green color and underline for the Reviewer #3, respectively, at exact locations on the revised manuscript. Besides, the manuscript has been edited by Michael D. Ash, a Canadian (blue color without underline). The revised manuscript was edited for proper English language, grammar, punctuation, spelling, and overall style by more of the high qualified native English language editors at Acceptediting Editing (Please find Supplementary Material for Review: English Editing Certificate). 

Reviewer #2

Comments and Suggestions for Authors

Authors prepared Eu-related compounds for luminescent shift.

Since they purchased those chemicals from a company. They would like to discuss over experimental conditions rather than chemical themselves. I strongly suggest they add more introduction to condition-related luminescent shift such as double-sided silica etc.

Ans:

  1. Thanks for your useful comments.
  2. In this study, three species of luminescent down-shifting (LDS) Eu-doped phosphors were purchased from InteMatix (Fremont, CA, USA). The average diameter of phosphors particles was about 15 μm. The thickness of p-type silicon substrate used in the work was 150 μm.
  3. In general, LDS and down conversion (DC) materials are used to convert high-energy incident photons into multiple photons (re-emitted) of lower energy to facilitate conversion into electricity for applications of photovoltaic devices. Typically, LDS and DC layers are suitable deposited on the front-side of solar cells. For double-sided, deposition of LDS and DC on front-side and up-conversion on rear-side of the solar cell may be a new approaches for harvesting high-energy and low-energy incident photons from the sun.
  4. For more introduction to condition-related luminescent shift, we cited two papers (Reference #19, #20), on the revised manuscript.

[19] Day J.; Senthilarasu S.; Mallick T.K. Improving spectral modification for

applications in solar cells: A review, Renew. Energy 2019, 132, 186-205.

DOI: 10.1016/j.renene.2018.07.101.

[20] Liu S.-M.; Chen W.; Wang Z.-G. Luminescence Nanocrystals for Solar Cell

Enhancement, J. Nanosci. Nanotechnol. 2010, 10, 1418-1429. DOI: 10.1166/jnn.2010.2023.

Currently, they discussed Eu-related compounds, where they did not design nor synthesize those chemicals for their purpose. For example, I would like to know what I can expect from regular reference chemical on silica-based substrate. I would like to know what the variables in the experiments are. If any functional groups or moiety of chemicals are important, they want to discuss over the moiety or functional groups on luminescent data. Currently, they just showed and summarized the results without approaches for each chemical’s properties. For example, they used tables 1 and 2 to summarize the conditions, but in the text, I do not see what they want to compare, analyze, and discuss. They just summarized the situations. Please give more detailed explanation for the purpose of the experiments. Thanks.

Ans:

  1. Yes, this study is focused on using the luminescent down-shifting spectral conversion to enhance efficiency of silicon solar cell via various species of Eu-doped phosphors and their combinations by spin-coating methods. This study is not focus on LDS material
  2. In this study, the LDS effects of various species of Eu-doped phosphors and their combinations were examined by PL, optical reflectance, and EQE spectra measurements. Based on PL measurement, the results indicated that the emission wavelength and intensity of LDS photons are depending on using species of Eu-doped phosphors and their combinations. The reflectance and EQE spectra of the samples with LDS layers are also used to check the LDS effects depending on the species of Eu-doped phosphors and their combinations. According to these results, we can evaluate how to choice a suitable Eu-doped phosphors species and their combinations for a silicon solar cell to achieve higher efficiency.
  3. Finally, we demonstrated and confirmed that LDS effects on improving efficiency of silicon solar cells using species of Eu-doped phosphors and their combinations by photovoltaic current density-voltage curves under AM 1.5G solar simulation. Impressive improvements in efficiency were observed for the cell with two-species LDS layers combination that of higher than single-species LDS layer, and single-species LDS layer was also higher than that of with a pure SiO2.

Reviewer 3 Report

In the manuscript "Characterization of luminescent down-shifting spectral conversion effects on silicon solar cells using various combinations of Eu-doped phosphors species", the authors presented a comparison study on single species and two species of Eu-doped phosphor as an antireflection coating applied to the front surface of silicon solar cells. The optical and electrical properties were studied.  I think this work is interesting. The double layer of SiO2 could efficiently reduce the front surface reflection by about 60% and increase the efficiency of the solar cell by about 30%. However, I have found the same authors' group several publications about the same topic, such as doi:10.3390/ma11050845, and doi.org/10.1016/j.tsf.2016.03.063. I am wondering what is new in this manuscript? There are many results are in common. Furthermore, I have a few comments that need to be addressed.

For the EDS measurements, why there are peaks for Ba, F, Sr, Ti, Mn, Cl? Does Fig. 4 for single-species samples or two-species combinations?

Same for Fig. 5 and 6. In the text and figure caption, it must be clarified whether it is for single-species samples or two-species.

Can the authors provide more discussion/details on why there is a shift in the PL peaks?

For Fig. 8, can the authors calculate the integrated Jsc from the EQE spectra? which I think it can summarize the overall outcome. For the figure caption and the related text, the EQE is called "spectra" rather than "response".

In line 249, the sentence "containing phosphors was higher that of the cell with only SiO2", I think "than" is missing before that.

Author Response

The authors would like to express their sincere appreciation to the Editors and Reviewers for useful comments. The reviewers’ comments are taken into account and mentioned issues have been addressed for significant improvement in the revised manuscript. In keeping with the comments of the Reviewer #1, Reviewer #2 and Reviewer #3, we have made some revisions by marking with the blue color and underline for the Reviewer #1, the red color and underline for the Reviewer #2, and the green color and underline for the Reviewer #3, respectively, at exact locations on the revised manuscript. Besides, the manuscript has been edited by Michael D. Ash, a Canadian (blue color without underline). The revised manuscript was edited for proper English language, grammar, punctuation, spelling, and overall style by more of the high qualified native English language editors at Acceptediting Editing (Please find Supplementary Material for Review: English Editing Certificate). The errors have been corrected at exact locations on the revised manuscript.

Reviewer #3

Comments and Suggestions for Authors

In the manuscript "Characterization of luminescent down-shifting spectral conversion effects on silicon solar cells using various combinations of Eu-doped phosphors species", the authors presented a comparison study on single species and two species of Eu-doped phosphor as an antireflection coating applied to the front surface of silicon solar cells. The optical and electrical properties were studied.  I think this work is interesting. The double layer of SiO2 could efficiently reduce the front surface reflection by about 60% and increase the efficiency of the solar cell by about 30%. However, I have found the same authors' group several publications about the same topic, such as doi:10.3390/ma11050845, and doi.org/10.1016/j.tsf.2016.03.063. I am wondering what is new in this manuscript? There are many results are in common.

Ans:

  1. Thanks for your useful comments.
  2. The previously reported on the Thin Solid Films journal in 2016 (doi:10.3390/ma11050845) was claimed that “performance enhancement of planar silicon solar cells through utilization of two luminescent down-shifting Eu-doped phosphor species”. This study was focused on an LDS layer comprising two species of europium (Eu)-doped phosphors in 3 wt% with emission wavelengths at 512 and 550 nm, 550 and 610 nm, and 512 and 610 nm, respectively. The improvement in efficiency of 15.97% for the cell with the combination of two Eu-doped phosphor species with emission wavelengths of 512 nm and 610 nm was obtained due to broad band luminescent emission and forward light scattering.
  3. The previously reported on the Materials journal in 2018 (doi:10.3390/ma11050845) was claimed that “photovoltaic performance enhancement of silicon solar cells based on combined ratios (0.5:1:1.5, 1:1:1, or 1.5:1:0.5) in 3 wt% of three species of europium-doped phosphors”. This study was focused on an LDS layer comprising three species of europium (Eu)-doped phosphors with various ratio and they mixed within a silicate solution using a spin-on film deposition. Impressive improvements in efficiency were observed in all three samples: 0.5:1:1.5 (20.43%), 1:1:1 (19.67%), 1.5:1:0.5 (16.81%) due to more broad band luminescent emission effects, compared to the control cell with a layer of pure SiO2.
  4. The present study, we focused on that two LDS layers (bottom LDS layer-the first coating/top LDS layer-the second coating) with different Eu-phosphors in the coating layer were deposited on the silicon solar cell by spin-on film method. The best improvement in efficiency of 31.63% was obtained when the cell deposited with the combination scheme of the first coating of Species-G and the second coating of Species-E, which also much higher than the efficiency of the previous reported.

Furthermore, I have a few comments that need to be addressed.

For the EDS measurements, why there are peaks for Ba, F, Sr, Ti, Mn, Cl? Does Fig. 4 for single-species samples or two-species combinations?

Ans:

  1. Yes, our EDS measurement results showed that the Eu-doped species samples presented small quantity of F, Ti, Mn, Mg, and Cl.
  2. Fig. 4 is the sample with single-species.

We revised as:

The composition of all phosphor samples was mainly Si, O, Ba, and Sr, with small quantity of Eu and other elements (F, Ti, Mn, Mg, and Cl)” and

Figure 4. EDS spectra of silicon substrates deposited a SiO2 layer containing a single phosphor: (a) Phosphor-E; (b) Phosphor-F; and (c) Phosphor-G”, on the revised manuscript.

Same for Fig. 5 and 6. In the text and figure caption, it must be clarified whether it is for single-species samples or two-species.

Ans:

Thanks for your remarked.

We revised as:

Figure 5. PL spectra and color optical emission images, for the samples with a SiO2 layer containing single phosphors: (a) Phosphor-E, (b) Phosphor-F, and (c) Phosphor-G”, and

Figure 6. PL spectra of samples with bottom/top LDS layers containing two phosphors: (a) E/F; (b) F/G; (c) G/E”, on the revised manuscript.

Can the authors provide more discussion/details on why there is a shift in the PL peaks?

Ans:

  1. For the samples with single-species Eu-doped phosphor, the PL emission peaks were 514.7 nm, 546.6 nm, and 603.3 nm for Species-A, Species-B, and Species-C, respectively.
  2. For the samples with two-species Eu-doped phosphors, in which the sample was deposited two LDS layers (bottom LDS layer-the first coating/top LDS layer-the second coating) with different Eu-phosphors in the coating layer. For example, Species-E (bottom LDS layer-the first coating)/Species-F (top LDS layer-the second coating), we noted as E/F combination. In general, the PL emission of two-species Eu-doped phosphors under excitation was the sum of PL emission intensity of the two species Eu-doped phosphors at the wavelengths. However, we found that the PL emission peak of the sample with E/F species was determined mainly by the top layer of Species-F because more laser lights excited on Species-F (top layer) than Species-E (bottom layer). Therefore, the PL emission peak of the sample with the combination of E/F species was at 545.5 nm which is near to the Species-F (546.6 nm). The wavelength shifted for F/G and G/E combinations also found the same as E/F combination.

For Fig. 8, can the authors calculate the integrated Jsc from the EQE spectra? which I think it can summarize the overall outcome. For the figure caption and the related text, the EQE is called "spectra" rather than "response".

Ans:

  1. Thanks very much for your remarked.
  2. We added the values of integrated Jsc from the EQE spectra were given in Table 1 for LDS and anti-reflection effects comparing.

We revised “EQE response” as “EQE spectra” on the figure caption and the related text.

We revised as:

Table 1 lists the integrated Jsc values from the EQE spectra to enable a comparison of LDS and anti-reflection effects”, on the revised manuscript.

In line 249, the sentence "containing phosphors was higher that of the cell with only SiO2", I think "than" is missing before that.

Ans:

Thanks very much for your remarked.

We revised as “The EQE of cells with a double coating of SiO2 containing phosphors was higher than that of the cell with only SiO2, except over a wavelength range of 460 to 630 nm, owing to LDS effects and forward scattering caused by the phosphors”, on the revised manuscript.

Round 2

Reviewer 3 Report

The authors have addressed the reviwer comments properly. I do not have further comments. Therefore, I recommend this manuscript to publish as is.